# An Implementation of LASER Beam Welding Simulation on Graphics Processing Unit Using CUDA

Ernandes Nascimento [1,*], Elisan Magalhães [1], Arthur Azevedo [1], Luiz E. S. Paes [2] and Ariel Oliveira [1]

1 Aeronautics Institute of Technology—ITA, São José dos Campos 12228-900, SP, Brazil; elisan@ita.br (E.M.); arthurama@ita.br (A.A.); arielafmo@ita.br (A.O.)
2 Faculty of Mechanical Engineering, Federal University of Uberlândia—UFU, Uberlândia 38410-337, MG, Brazil; luiz.paes@ufu.br
* Correspondence: ernandes@ita.br

**Abstract:** The maximum number of parallel threads in traditional CFD solutions is limited by the Central Processing Unit (CPU) capacity, which is lower than the capabilities of a modern Graphics Processing Unit (GPU). In this context, the GPU allows for simultaneous processing of several parallel threads with double-precision floating-point formatting. The present study was focused on evaluating the advantages and drawbacks of implementing LASER Beam Welding (LBW) simulations using the CUDA platform. The performance of the developed code was compared to that of three top-rated commercial codes executed on the CPU. The unsteady three-dimensional heat conduction Partial Differential Equation (PDE) was discretized in space and time using the Finite Volume Method (FVM). The Volumetric Thermal Capacitor (VTC) approach was employed to model the melting-solidification. The GPU solutions were computed using a CUDA-C language in-house code, running on a Gigabyte Nvidia GeForce RTX™ 3090 video card and an MSI 4090 video card (both made in Hsinchu, Taiwan), each with 24 GB of memory. The commercial solutions were executed on an Intel® Core™ i9-12900KF CPU (made in Hillsboro, Oregon, United States of America) with a 3.6 GHz base clock and 16 cores. The results demonstrated that GPU and CPU processing achieve similar precision, but the GPU solution exhibited significantly faster speeds and greater power efficiency, resulting in speed-ups ranging from 75.6 to 1351.2 times compared to the CPU solutions. The in-house code also demonstrated optimized memory usage, with an average of 3.86 times less RAM utilization. Therefore, adopting parallelized algorithms run on GPU can lead to reduced CFD computational costs compared to traditional codes while maintaining high accuracy.

**Keywords:** Nvidia CUDA®; CUDA-C code; GPU processing; finite volume method; LASER beam welding





## 1. Introduction

Numerical techniques to find approximated solutions instead of exact ones were found to be helpful in various fields of science, engineering, physics, and many other disciplines where mathematical models were used to describe real-world phenomena. Over the past 60 years, as noted by Thomée V. [1], the research and development of computational methods have successfully addressed numerous engineering problems, including heat transfer and fluid flow. As a result, the field of Computational Fluid Dynamics (CFD) became an essential part of the modern industrial design process [2].

Numerical algorithms involve iterative processes, where an initial guess is refined through successive calculations to approach the true solution. The process continues until a predefined convergence criterion is met. Still, the processing efficiency and solution accuracy are strongly related to the hardware capacity and software optimization. For instance, the electronic connections between processor and memory units may limit the data throughput. Therefore, it is important to evaluate the Random Access Memory (RAM) and Video Random Access Memory (VRAM) implementations in terms of operational

speeds [GHz] [3]. The RAM device may be defined as the main computer memory used to store and process data, being placed at the computer's motherboard. Conversely, GPUs possess a distinct and non-removable type of memory known as VRAM, which is directly integrated into the graphics card. In GPU processing platforms such as CUDA®, RAM is often referred to as host machine memory, while VRAM is termed device memory [4].

Over the past few decades, many commercial CFD packages have been developed. However, most are designed to perform computations based on CPU processors. With the advent of high-performance computing, parallelized numerical methods have become increasingly important. When parallel computing is employed, calculations are distributed among multiple processors or cores to efficiently solve large-scale problems. In such conditions, the computational cost is often decreased compared to sequential computation via CPUs.

A subject of industrial interest is yield stress fluids, which do not deform until the yield stress is exceeded. In the case of flow into a narrow eccentric annulus, this type of phenomenon can be decomposed into multiple long-thin flows. The nonlinearity in the governing equations requires substantial calculations, so the Lagrangian algorithm is often applied. Medina Lino et al. (2023) [5] proposed implementing a non-Newtonian Hele–Shaw flow to model the displacement of Herschel–Bulkley fluids in narrow eccentric annuli. They utilized the CUDA® Fortran language to accelerate calculations compared to CPU processing. The calculations run in an NVIDIA GeForce® RTX™ 2080 Ti were up to 40 times faster than the simulations run in an Intel® Core™ I7 3770 processor.

Continuing in the field of fluid flow modeling, Xia et al. (2020) [6] developed a CUDA-C language GPU-accelerated package for simulation of flow in nanoporous source rocks with many-body dissipative particle dynamics. The authors demonstrated through a flow simulation in realistic shale pores that the CPU counterpart requires 840 Power9 cores to rival the performance delivered by the developed package with only four Nvidia V100 GPUs. More recently, Viola et al. (2022) [7] applied CUDA to perform GPU-accelerated simulations of the Fluid–structure–electrophysiology Interaction (FSEI) in the left heart. The resulting GPU-accelerated code can solve a single heartbeat within a few hours (ranging from three to ten hours depending on the grid resolution), running on a premises computing facility consisting of a few GPU cards. These cards can be easily installed in a medical laboratory or hospital, thereby paving the way for a systematic Computational Fluid Dynamics (CFD)-aided diagnostic approach.

Simulations in the field of computational biomedicine have also been accelerated with the aid of GPU processing. The desire to create a three-dimensional virtual human as a digital twin of one's physiology has led to the development of simulations using the CUDA® computing platform as a means of reducing processing time. For example, the HemeLB solver, which is based on the lattice Boltzmann method, is widely utilized for simulating blood flow using real patient images. Zacharoudiou et al. (2023) [8] utilized the method's strong scaling capability to adapt their algorithm for execution on a GPU architecture using CUDA-C language. Indeed, such scalability extends to a higher level of parallelism for GPU codes compared to CPU codes. When comparing computations using an equivalent number of GPU and CPU threads, computations using the GPU were still up to 85 times faster. The authors compared different settings of supercomputers.

Applying the GPU for calculations may also facilitate the achievement of more detailed and realistic simulations. In 2021, O'Connor and Rogers [9] adapted and implemented the open-source DualSPHysics code to run on a GPU. This adaptation was aimed at achieving more reliable simulations of coupled interactions between free-surface flows and flexible structures, addressing concerns that frequent use of reduced models may lead to erroneous assumptions. The execution time needed to perform the calculations using an NVIDIA™ Tesla® V100 GPU and an Intel® Xeon™ E5 2690 were compared. The GPU outperformed the CPU for all numbers of particles investigated. However, the speed-up was proportional to the number of particles. Thus, when dealing with a small number of particles, the speed-up

on the GPU was relatively low. As the number of particles increased, so did the speed-up, reaching up to 50 times faster on the GPU.

In addition to fluid flow, some authors also use numerical models computed through GPUs to investigate heat transfer. For example, Satake et al. (2012) [10] performed optimizations of a GPU-accelerated heat conduction equation by a programming of CUDA Fortran from an analysis of a Parallel Thread Execution (PTX) file. Before implementing the proposed code corrections, CUDA-C exhibited a speed 1.5 times faster than by CUDA Fortran. Later, Klimeš and Štětina (2015) [11] employed the Finite Difference Method (FDM) to perform three-dimensional simulations with solidification modeling. The results demonstrated that the GPU implementation outperformed CPU-based simulations by 33–68 times when utilizing a single Nvidia Tesla C2075 GPU to execute kernels. This considerable speed-up was enough to enable the application of their method in real-time scenarios. Szénási (2017) [12] solved the Inverse Heat Conduction Problem (IHCP) using NVLink capable power architecture between the host and devices. This implementation (running on four GPUs) was about 120 times faster than a traditional CPU implementation using 20 cores.

Continuing the literature review in GPU-based computational methods in heat transfer, Semenenko et al. (2020) [13] simulated conductive stationary heat transfer on a two-dimensional domain to compare the performance of CPU and GPU architectures. Their study was performed through several simulations using various hardware configurations, including four different GPUs: AMD Radeon™ RX VEGA® 56, NVIDIA GeForce® GTX™ 1060, NVIDIA GeForce® GTX™ 860 m, and NVIDIA Tesla™ M40®. It also utilized five Intel® Core™ i7 CPU processors: 3630 QM, 4720 HQ, 6700 K, 7700, and 7820 HQ. Different numbers of mesh elements were simulated. The results indicated that with an increase in the number of elements in the mesh, GPU calculations were faster compared to those on the CPU. Across all configurations considered, the GPU was, on average, 9 to 11 times faster than the CPU.

Convective and radiative heat transfers can also be studied using parallel computing. For instance, Taghavi et al. (2021) [14] performed simulations of convective heat transfer in nanofluids inside a sinusoidal wavy channel. The authors solved the tridiagonal matrices obtained through the Spline Alternating Direction Implicit (SADI) technique using the Parallel Thomas Algorithm (PTA) on the GPU and the classic Thomas algorithm on the CPU, respectively. Implementing this high-order method on the GPU significantly reduced the computing time. The simulations could be performed up to 18.32 times faster on a GeForce® GTX™ 970 than on an Intel® Core™ i7 5930K processor. Additionally, the Monte Carlo, Runge–Kutta, and ray tracing methods were combined to simulate radiative heat transfer in a graded-index (GRIN) medium. Despite providing high precision, such sequential computations often require a significant amount of computational time.

Shao et al. (2021) [15] developed two- and three-dimensional models optimized for graded-index (GRIN) media using parallel computing on GPUs to enhance processing. Computational times were compared between GPU implementations using an NVIDIA GeForce® GTX™ 1080 Ti and CPU implementations using the Intel® Core™ i7 8750H and the Xeon™ Gold 5120 processors. In the two-dimensional model, the GPU demonstrated a speed-up of over 43 times and 5 times compared to the equivalent CPU implementation using a single core and six CPU cores, respectively. In the three-dimensional case, the GPU was 35 times and 2 times faster than the CPU, considering a single core and 14 CPU cores sequentially.

As discussed in the previous literature review, crucial contributions have made possible the acceleration of computational processing times through the application of GPU parallelization in several fields of engineering research. The CPU-based processing has been investigated for over fifty years, while GPU processing methods are still in early development, having been a research focus for nearly fifteen years. Hence, there is ample opportunity to explore new parallelized methods and specific studies for better addressing the GPU capabilities. For instance, in previous work, Azevedo et al. (2022) [16] compared nonlinear and constant thermal properties approaches applied for estimating the

temperature in LASER Beam Welding (LBW) simulations. The authors conducted a detailed study on the temperature gradient, its influence on thermocouple positioning, and a methodology to evaluate thermal properties convergence. However, the results were not extensively compared in terms of processing performance, energy, optimization and accuracy to well-established commercial code solutions. Therefore, in the present research, the advantages and drawbacks of an implementation of LASER Beam Welding (LBW) simulation using CUDA were investigated. The developed numerical solution utilizes CPU and GPU runtime code functions, along with multithreaded GPU parallelization, as illustrated in Figure 1.

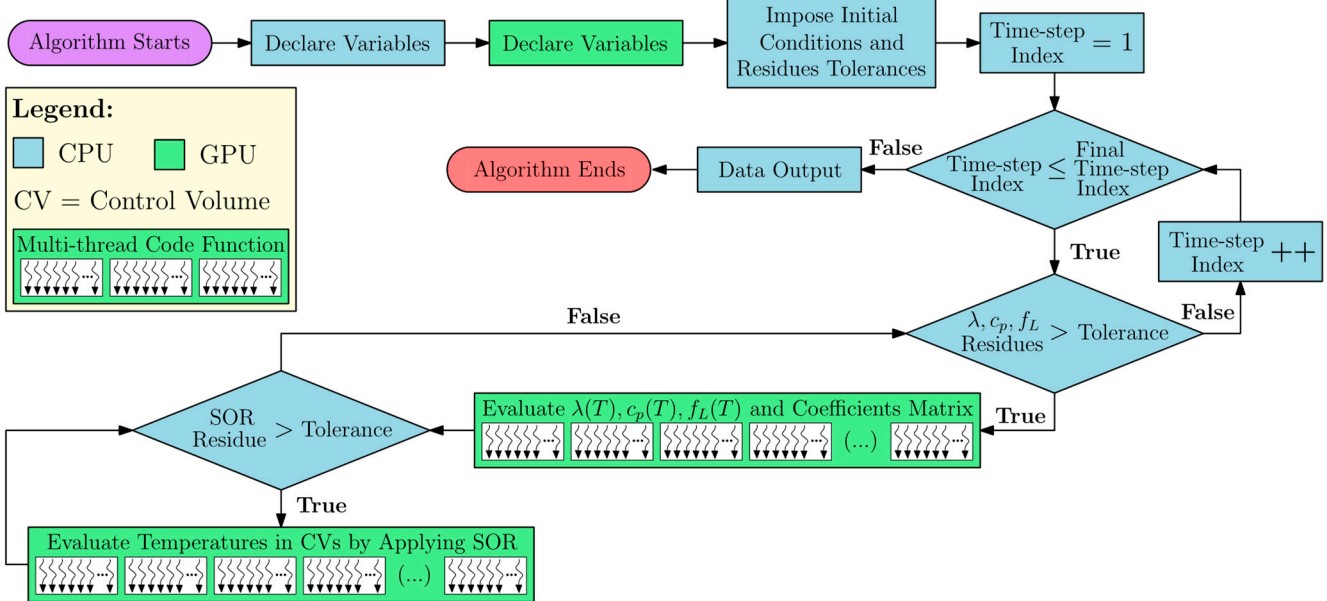

**Figure 1.** Developed CUDA-C language code flowchart.

The investigation was conducted through the application of the heat conduction Partial Differential Equation (PDE) with a transient total enthalpy term to model the LBW process. The latter is needed to account for the phase change (melting) according to the Volumetric Thermal Capacitor (VTC) approach [17]. The equations were discretized in space and time over a three-dimensional domain by applying the Finite Volume Method (FVM). The heat losses through convection at the boundaries of the domain were accounted for using Newton's law of cooling and the losses through radiation were calculated by applying the Stefan–Boltzmann law. A Gaussian conical profile models the welding heat source. The effects of implementing constant and temperature-dependent thermophysical properties for the specimen's material were evaluated. The GPUs simulations were performed in an in-house code written in CUDA-C language and run in an Nvidia™ Geforce® RTX™ 3090 and a Geforce® RTX™ 4090, both with 24 GB of video memory. A parallelized form of the Successive Over-Relaxation (SOR) solver was used to find the solution of the linear system of equations. The CUDA® code, as well as the three other top-rated CPU-based commercial codes, were executed on a desktop PC equipped with an Intel® Core™ i9 12900KF processor. The temperature profiles simulated using equivalent solutions produced by GPU and CPU were compared, as well as the computational performance in terms of processing time, energy consumption, cost efficiency, and memory usage. The enhanced performance demonstrated in the research results highlights the significant potential for GPUs to replace CPUs in CFD applications.

## 2. Materials and Methods

### 2.1. The Laser Beam Welding (LBW) Simulation

The Laser Beam Welding (LBW) technique is a high-precision process that makes use of a concentrated light beam to join metals together. The method yields high-quality welds due to a low Heat Affected Zone (HAZ) resultant from the high precision of the laser and the accurate control over the welding parameters, thus minimizing distortions and retaining much of the material's original mechanical properties. The technique also offers easy automation, which results in a high-speed manufacturing process, as the laser generates enough heat to rapidly move across the workpiece. The positive final characteristics of the LBW joint and its versatility contributed to a significant increase in the method's popularity in the last few years. For instance, LBW is currently by far the most simulated welding technique present in recent scientific publications [18]. The previous simulation of the process allows for various advantages such as optimization of the technique through new modeling and parameters tuning [19], enhanced materials selection [20], the prediction of the final weld bead mechanical characteristics [21,22], and the estimation of involved parameters through inverse analysis [23–25]. Hence, the present computational performance analysis was performed by simulating an LBW process conducted by an automated LASER head focused on an SAE 1020 steel specimen. The welding process and its geometrical parameters and thermophysical considerations are schematized in Figure 2, where $L_x$, $L_y$, and $L_z$ are the specimen lengths at $x$, $y$, and $z$ directions, respectively, $L_w$ is the weld bead length, $u$ is the LASER head velocity, $T_\infty$ is the ambient temperature, $q''_{rad}$ is the rate of heat loss by radiation per unit area, and $q''_{con}$ is the rate of heat loss by convection per unit area. The values for each LBW parameter are tabulated in Table 1.

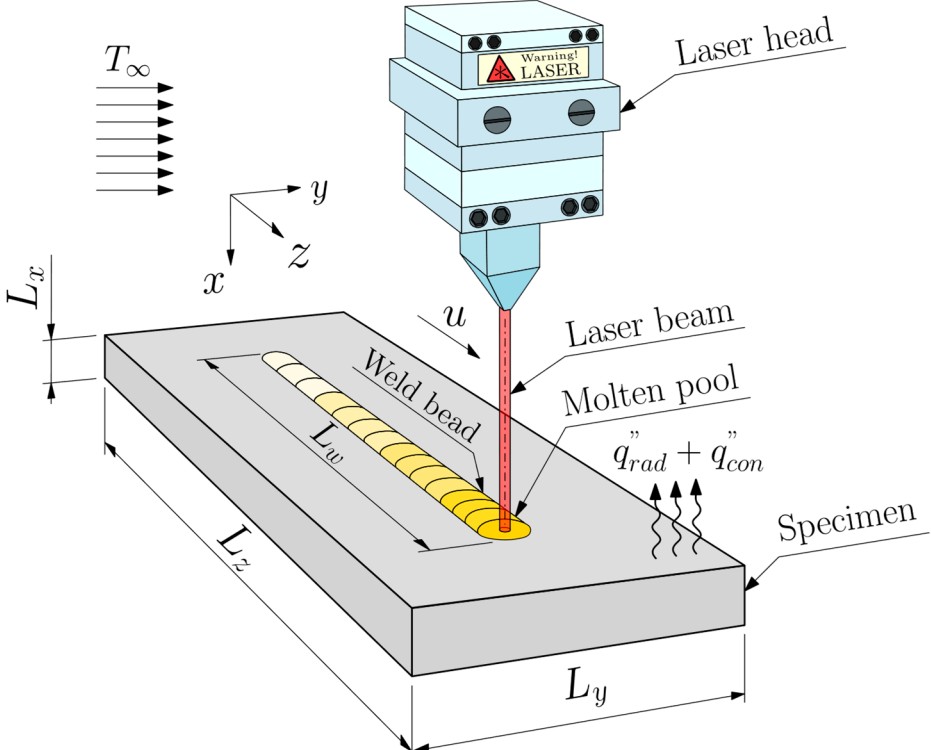

**Figure 2.** Simulated LASER Beam Welding (LBW) process schematics, parameters, and thermal modeling.

**Table 1.** LBW process parameters.

| Parameter | Values |
|---|---|
| Ambient temperature ($T_\infty$) [°C] | 20.0 |
| LASER head velocity ($u$) [mm/min] | 3000.0 |

**Table 1.** *Cont.*

| Parameter | Values |
|---|---|
| Sample length at $x$-direction ($L_x$) [mm] | 9.5 |
| Sample length at $y$-direction ($L_y$) [mm] | 20.0 |
| Sample length at $z$-direction ($L_z$) [mm] | 40.0 |
| Weld bead length ($L_w$) [mm] | 30.0 |

*2.2. Mathematical Model*

A transient three-dimensional heat conduction equation physically governs the simulated welding process. The phenomenon may be modeled through a PDE with a volumetric heat generation term aimed at quantifying the heat input and a transient term written as a function of the total enthalpy to account for the metal phase change. Hence, the final governing equation may be written as follows [26]:

$$\underbrace{\frac{\partial}{\partial x}\left(\lambda\frac{\partial T}{\partial x}\right) + \frac{\partial}{\partial y}\left(\lambda\frac{\partial T}{\partial y}\right) + \frac{\partial}{\partial z}\left(\lambda\frac{\partial T}{\partial z}\right) + \dot{g}}_{\text{Three−dimensional heat conduction}} = \underbrace{\frac{\partial H}{\partial t}}_{\substack{\text{Transient} \\ \text{term}}} \tag{1}$$

where $x$, $y$, and $z$ are the cartesian coordinates, $\lambda$ is the nonlinear thermal conductivity, $T$ is the temperature, $\dot{g}$ is the volumetric heat source rate, and $t$ is the physical time. The total enthalpy term, $H$, can be mathematically written as [27]:

$$H = \underbrace{\rho\int_0^T c_p(\varphi)d\varphi}_{\substack{\text{Portion related to} \\ \text{the sensible heat}}} + \underbrace{\rho f_L(T)L_f}_{\substack{\text{Portion related to} \\ \text{the latent heat}}} \tag{2}$$

where $\rho$ is the density, $c_p$ is the specific heat at constant pressure, $f_L$ is the temperature-dependent liquid mass fraction function, $\varphi$ is the generic integration variable, and $L_f$ is the latent heat of fusion. The total enthalpy term was solved through a partial implementation of the Volumetric Thermal Capacitor (VTC) method [17]. A linear temperature-dependent function was applied to model the materials' fusion. This equation may be written as follows [25]:

$$f_L(T) = \begin{cases} 0 \text{ if } T < T_m \\ 1 \text{ if } T > T_m \end{cases}, 0 < f_L < 1 \text{ if } T = T_m \tag{3}$$

where $T_m$ is the melting temperature. The heat losses by convection and radiation were calculated based on Newton's law of cooling and the Stefan–Boltzmann law, respectively. The final heat loss equation may be written as follows [28]:

$$q''_L = -\lambda\frac{\partial T}{\partial \eta} = \underbrace{h(T)(T - T_\infty)}_{\substack{\text{Newton's law} \\ \text{of cooling}}} + \underbrace{\sigma\phi_{rad}(T)\left(T^4 - T_\infty^4\right)}_{\text{Stefan Boltzmann law}} \tag{4}$$

where $\eta$ is the direction normal to the surface, $h(T)$ is the temperature-dependent convection heat transfer coefficient, $\sigma$ is the Stefan–Boltzmann constant, and $\phi_{rad}$ is the material's emissivity.

### 2.3. Moving Heat Source

The automated LASER head heat source was modeled as a constant velocity mobile Gaussian whole conical volumetric profile, implemented here as tuned, and reviewed in previous work [18,29]. The heat distribution may be mathematically written as

$$\dot{g} = \frac{Q_w}{0.460251 h_p R^2} e^{-\frac{4.5(z-ut)^2}{R^2}} e^{-\frac{4.5(y-L_y/2)^2}{R^2}} \left(1 - \frac{x^{1/2}}{h_p^{1/2}}\right) \tag{5}$$

where $Q_w$ is the LASER heat source power, $h_p$ is the height of penetration, $R$ is the welding radius, and $u$ is the LASER head velocity. The applied heat source model and its geometrical parameters are schematized in Figure 3.

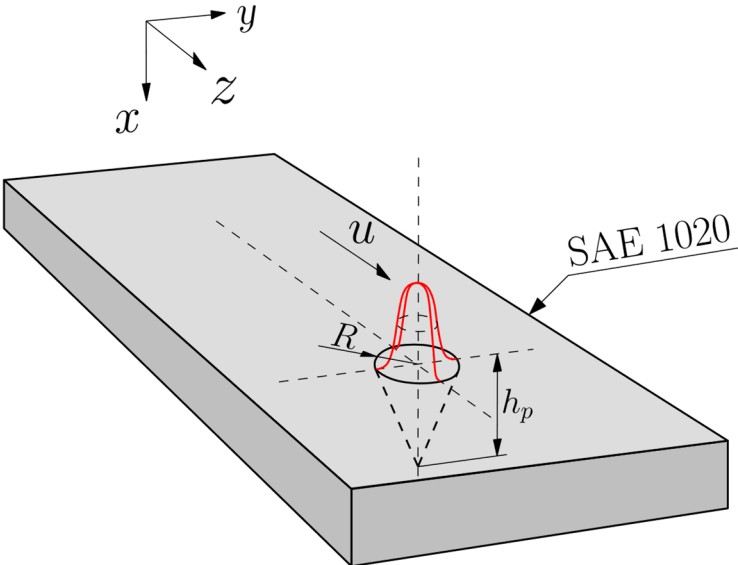

**Figure 3.** Gaussian whole conical volumetric heat source profile.

The heat source model parameters are tabulated in Table 2.

**Table 2.** Gaussian whole conical heat source model parameters.

| Parameter | Values |
|---|---|
| Height of penetration ($h_p$) [mm] | 1.65 |
| LASER power ($Q_w$) [W] | 800.0, 1200.0 |
| Welding radius ($R$) [mm] | 0.5 |

### 2.4. Post Processing, Spatial and Temporal Meshes Independencies

The four probe points, $P_1$ to $P_4$, were positioned transversally to the weld bead to allow for measurement and comparison of the resultant temperature fields simulated by each code run. Instead of parallel to the welding direction, the transverse positioning of the probe points allows for an enhanced numerical convergence analysis by avoiding a similar shape to all the curves. This alternative displacement results in a different temperature magnitude of the curves as well as the different peak times caused by the thermal inertia variance resultant from the smaller ($P_1$) and larger distances ($P_3$ and $P_4$) between the reading points and the heat core. The cartesian coordinates for each probe point are presented in Table 3 and its positions in the specimen are illustrated in Figure 4.

**Table 3.** Probe points three-dimensional cartesian coordinates.

| Coordinates | $P_1$ | $P_2$ | $P_3$ | $P_4$ |
|---|---|---|---|---|
| $x$ [mm] | 9.5 | 9.5 | 9.5 | 9.5 |
| $y$ [mm] | 10.0 | 11.0 | 12.0 | 13.0 |
| $z$ [mm] | 20.0 | 20.0 | 20.0 | 20.0 |

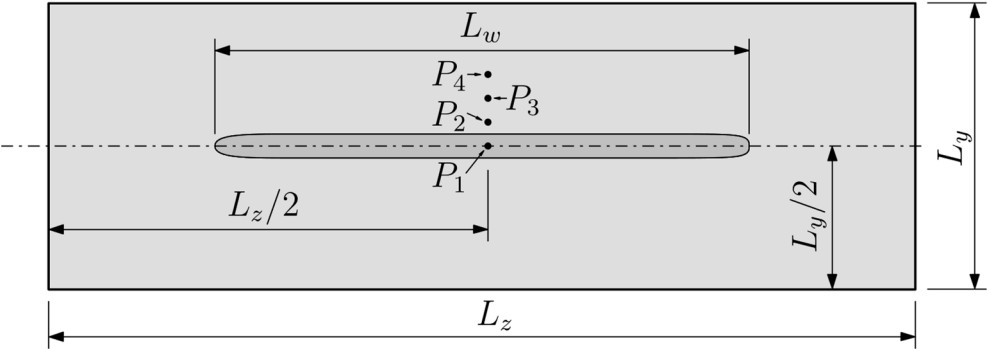

**Figure 4.** Applied temperature probe points ($P_i$) positioning.

A mesh size independence study was performed to ensure results independent of spatial refinement. The simulated domain was built as a uniform orthogonal structured grid. The resultant temperature fields became independent of mesh refinement at nearly 3,000,000 total nodes, distributed in nearly equal proportion to the specimen lengths at each dimension. Hence, a final mesh with 3,203,904 total nodes was instead used ($82 \times 148 \times 264$ nodes, at $x$, $y$, and $z$, respectively), resulting in an added design safety factor of approximately 6.797%. The average absolute errors ($E_{avg}$) [%] between a very refined mesh ($G_7$) and the other investigated cases ($G_n$) are tabulated in Table 4, where $N_x$, $N_y$, and $N_z$ are the number of nodes at $x$, $y$, and $z$ directions, respectively, and $N_T$ is the total number of nodes.

**Table 4.** Average absolute error ($E_{avg}$) [%] between mesh size $G_7$ and others for probe points $P_1$ and $P_2$.

| Mesh ($G_n$) | $N_x \times N_y \times N_z$ = Total Nodes ($N_T$) | $P_1$—Error ($E_{avg}$) [%] | $P_2$—Error ($E_{avg}$) [%] |
|---|---|---|---|
| $G_1$ | $24 \times 48 \times 82 = 94{,}464$ | 27.608 | 25.787 |
| $G_2$ | $9 \times 58 \times 113 = 190{,}066$ | 3.867 | 3.642 |
| $G_3$ | $37 \times 74 \times 137 = 375{,}106$ | 3.296 | 3.792 |
| $G_4$ | $46 \times 95 \times 177 = 773{,}490$ | 0.785 | 0.812 |
| $G_5$ | $62 \times 126 \times 208 = 1{,}624{,}896$ | 1.811 | 1.540 |
| $G_6$ | $82 \times 148 \times 264 = 3{,}203{,}904$ | 0.029 | 0.157 |
| $G_7$ | $111 \times 172 \times 325 = 6{,}204{,}900$ | - | - |

A time-step size independence analysis was also conducted to investigate the temperature fields dependency on the temporal grid refinement. The results became independent of time-step size for values smaller than $2.5 \times 10^{-3}$ [s]. However, the values of $1.0 \times 10^{-3}$ and $2.5 \times 10^{-5}$ s were instead applied. These values were intercalated, depending on the combination of input parameters, in direct proportion to the applied heat source power [W]. Time-step size values larger than $2.5 \times 10^{-5}$ [s] will result in solver failure for some of the commercial codes in the analysis when a 1200 [W] heat source power or more is applied.

### 2.5. Material Properties

The welded material in the simulated LBW process is the SAE 1020 steel. Variations between constant and temperature-dependent thermal properties were applied for the specific heat ($c_p$) and the thermal conductivity ($\lambda$) to verify the quality of the CUDA® in-house code resultant data in both cases. The corresponding values of the properties at 20 °C were fixed and applied whenever constant properties were used. The thermophysical properties of

the SAE 1020 steel specimen are exposed in Table 5. The temperature-dependent behaviors of the specific heat ($c_p$) (input units in Kelvin) and the thermal conductivity ($\lambda$) (input units in Celsius degrees) are depicted in Figure 5.

**Table 5.** SAE 1020 steel alloy thermophysical properties.

| Thermal Properties (SAE 1020) | Values/Equations |
|---|---|
| Density ($\rho$) [kg/m$^3$] | 7731.3 |
| Emissivity ($\phi_{rad}$) | 0.8 |
| Latent heat of fusion ($L$) [kJ/kg] | 247.0 |
| Melting temperature ($T_m$) [°C] | 1450.0 |
| Specific heat ($c_p$) [J/kg·K] | $c_p(T) = 3.298 \times 10^2 e^{1.509 \times 10^{-3} T}$ |
| Thermal conductivity ($\lambda$) [W/m·K] | $\lambda(T) = 2.5 \times 10^{-5} T^2 - 0.053T + 57.2$ |

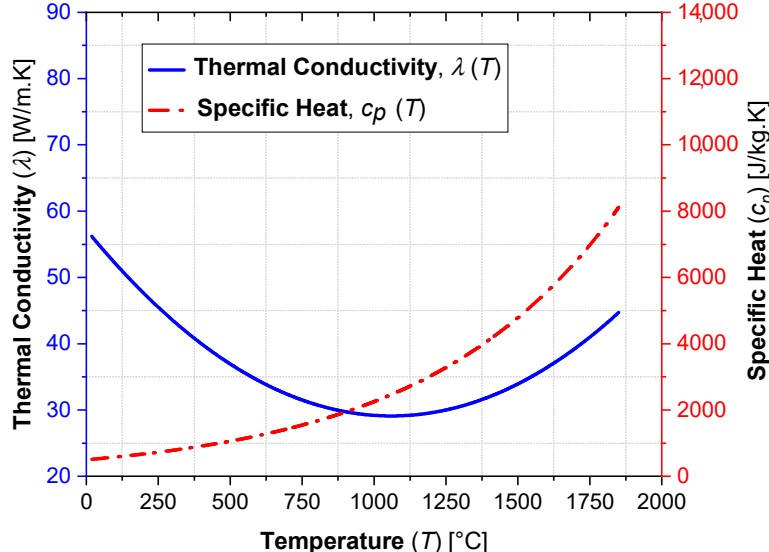

**Figure 5.** Specific heat ($c_p$) [J/kg·K] and thermal conductivity ($\lambda$) [W/m·K] behavior as a function of temperature ($T$) [°C].

### 2.6. Simulation Parameters

Despite all the commercial code solutions using multigrid techniques, the Successive Over-relaxation (SOR) parallelized solver was applied in the GPU solution without using multigrid techniques. All the simulations were performed with a first-order scheme temporal discretization. An energy residual convergence criterion was applied to all cases, and the threshold was set to $1.0 \times 10^{-5}$. Finally, the simulation parameters are presented in Table 6.

**Table 6.** Simulated cases parameters.

| Parameter | Values |
|---|---|
| Temporal discretization | First-order scheme |
| Time-step ($\Delta t$) [s] | $1.0 \times 10^{-3}$ and $2.5 \times 10^{-5}$ |
| Simulation total time ($t_{Tot}$) [s] | 2.5 |
| Solver convergence criterion | Energy residual |
| Residual threshold | $1.0 \times 10^{-5}$ |

## 3. Results and Discussion

### 3.1. LBW Temperature Fields

The first analysis was conducted with the aim of verifying the accuracy of the GPU solution and its code behavior when constant or temperature-dependent thermal properties

are applied. As there is a notable drop in thermal conductivity ($\lambda$) values when the temperature increases (as in Figure 5), the LASER power ($Q_w$) was firstly reduced to 800.0 [W] to avoid excessively high temperatures when constant properties are applied. The curves presented in Figure 6 illustrate the temperature fields for the CUDA® code and commercial solutions through probe points $P_1$ to $P_4$. The average absolute error ($E_{avg}$) [%] between the GPU code and each commercial solution is included in Table 7. The maximum error reached was 3.746% between the CUDA® code and the commercial code #2 for probe point $P_3$. However, the equivalent measurements probed at the other commercial codes returned values with much better agreement.

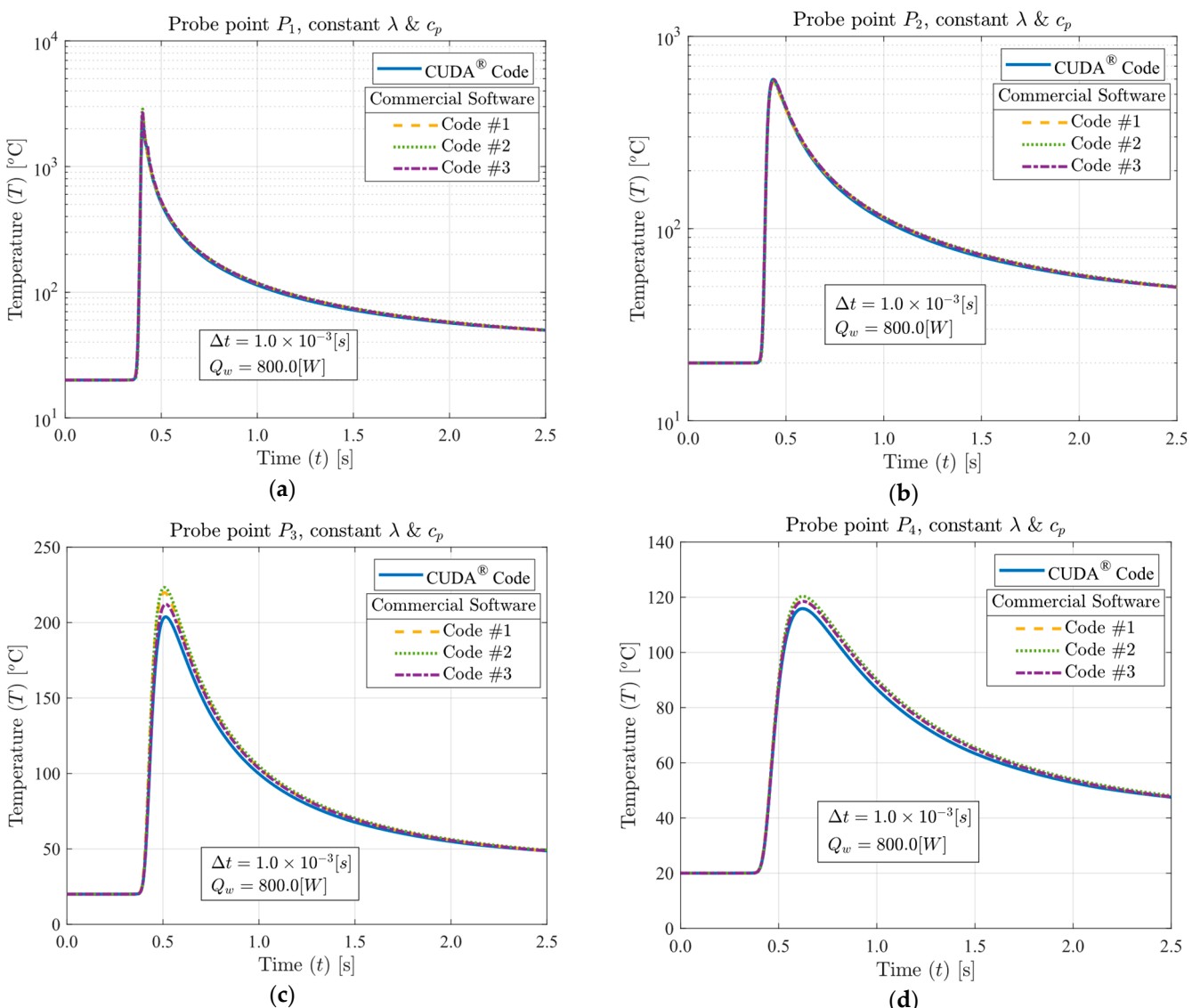

**Figure 6.** Temperature ($T$) [°C] as a function of time ($t$) [s] for 800 [W] LASER power ($Q_w$), $1.0 \times 10^{-3}$ [s] time-step size ($\Delta t$), constant $\lambda$, constant $c_p$ and (**a**) Probe point $P_1$, (**b**) Probe point $P_2$, (**c**) Probe point $P_3$, (**d**) Probe point $P_4$.

**Table 7.** Average absolute error ($E_{avg}$) [%] between CUDA® and each processing code (#$n$) as a function of probe point ($P_n$) for constant $\lambda$ and constant $c_p$.

| Probe Point/Solution | Average Absolute Error ($E_{avg}$) [%] | | |
| --- | --- | --- | --- |
| | Commercial Code #1 | Commercial Code #2 | Commercial Code #3 |
| $P_1$ | 2.070 | 3.051 | 2.180 |
| $P_2$ | 1.530 | 2.456 | 1.810 |
| $P_3$ | 2.843 | 3.746 | 2.065 |
| $P_4$ | 1.678 | 2.582 | 1.575 |

Applying temperature-dependent thermal properties caused the temperatures at probe point $P_1$ to drop nearly 55.6%. The resultant temperature fields are illustrated in Figure 7. The overall good matching between the GPU and the commercial solutions was kept, and the maximum average absolute error was 2.756% at probe point $P_3$ for CUDA® and commercial code #2. The average absolute error ($E_{avg}$) values are presented in Table 8. The first two sets of results evidenced that the higher the thermal gradient achieved, the higher the errors involved in all code solutions.

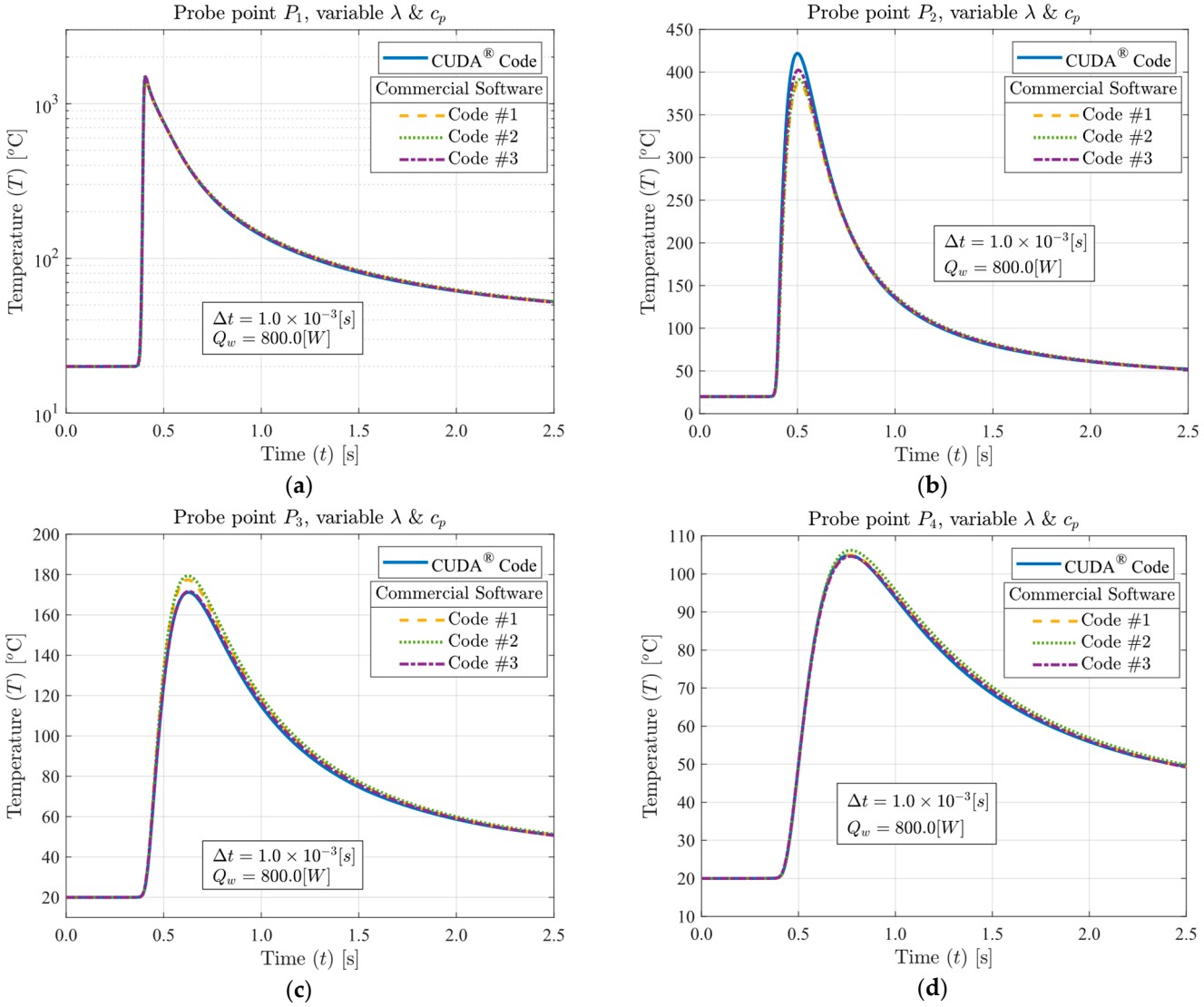

**Figure 7.** Temperature ($T$) [°C] as a function of time ($t$) [s] for 800 [W] LASER power ($Q_w$), $1.0 \times 10^{-3}$ [s] time-step size ($\Delta t$), variable $\lambda$, variable $c_p$ and (**a**) Probe point $P_1$, (**b**) Probe point $P_2$, (**c**) Probe point $P_3$, (**d**) Probe point $P_4$.

**Table 8.** Average absolute error ($E_{avg}$) [%] between CUDA® and each processing code (#$n$) as a function of probe point ($P_n$) for variable $\lambda$, variable $c_p$ and 800 [W] LASER power ($Q_w$).

| Probe Point/Solution | Average Absolute Error ($E_{avg}$) [%] | | |
| :---: | :---: | :---: | :---: |
| | Commercial Code #1 | Commercial Code #2 | Commercial Code #3 |
| $P_1$ | 1.059 | 2.149 | 0.929 |
| $P_2$ | 1.475 | 2.280 | 1.092 |
| $P_3$ | 1.768 | 2.756 | 0.863 |
| $P_4$ | 0.635 | 1.521 | 0.600 |

The temperature field investigation was finalized by increasing the LASER power ($Q_w$) to 1200.0 [W] and applying temperature-dependent thermal properties to better address the precision of the applied phase change modeling. For the sake of brevity, only the temperature curves for probe points $P_1$ and $P_2$ were plotted, as in Figure 8. Here, the maximum absolute average error ($E_{avg}$) observed was 4.617% at probe point $P_3$ for commercial code #2. The calculated error values are presented in Table 9. As expected, the increase in thermal gradient magnitude resulted in higher error values at probe point $P_1$ compared to the errors presented in Table 8.

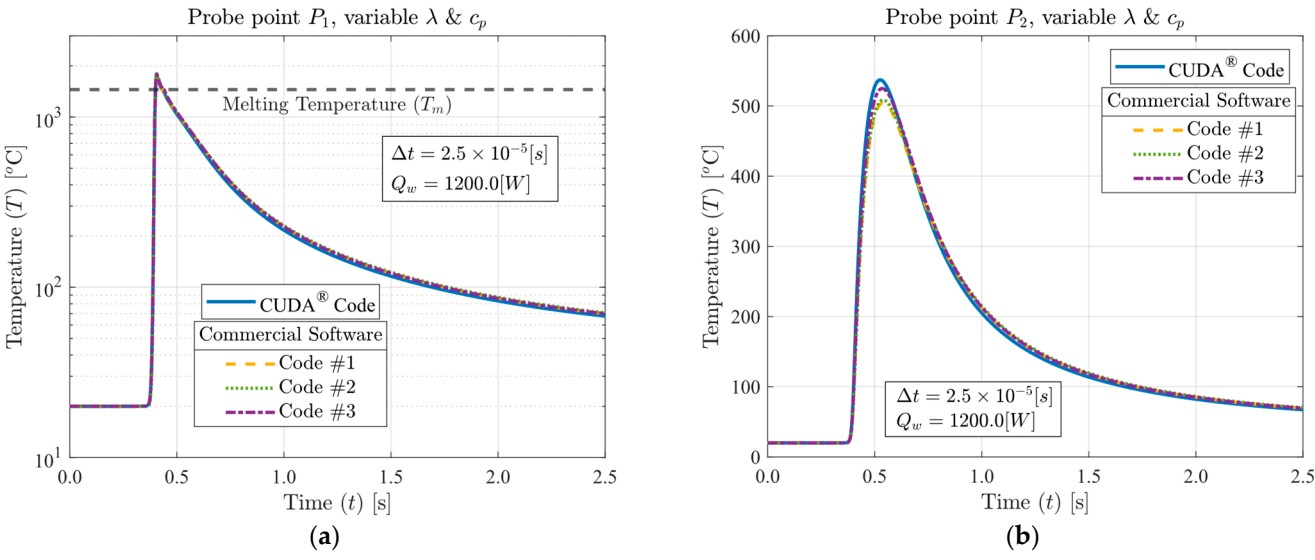

**(a)**              **(b)**

**Figure 8.** Temperature ($T$) [°C] as a function of time ($t$) [s] for 1200 [W] LASER power ($Q_w$), $2.5 \times 10^{-5}$ [s] time-step size ($\Delta t$), variable $\lambda$, variable $c_p$ and (**a**) Probe point $P_1$, (**b**) Probe point $P_2$.

**Table 9.** Average absolute error ($E_{avg}$) [%] between CUDA® and each processing code (#$n$) as a function of probe point ($P_n$) for variable $\lambda$, variable $c_p$ and 1200 [W] LASER power ($Q_w$).

| Probe Point/Solution | Average Absolute Error ($E_{avg}$) [%] | | |
| :---: | :---: | :---: | :---: |
| | Commercial Code #1 | Commercial Code #2 | Commercial Code #3 |
| $P_1$ | 3.291 | 4.615 | 2.539 |
| $P_2$ | 3.076 | 4.248 | 1.976 |
| $P_3$ | 3.497 | 4.617 | 1.117 |
| $P_4$ | 1.996 | 2.937 | 1.487 |

### 3.2. CUDA® vs. Variable CPU Processing Nodes

The investigation was continued by analyzing the change in computational performance of the commercial solutions through the variation of the number of CPU parallel processing nodes ($N_p$). The consumer-available codes were set to run in three, five, and seven CPU processing nodes. The CUDA® code is a GPU parallelized solution; thus, it runs on only one CPU processing node. The overall performance of all solutions is included

in the bar charts of Figure 9 for an LBW simulated case with 800 [W] LASER power ($Q_w$), $1.0 \times 10^{-3}$ [s] time-step size ($\Delta t$), variable $\lambda$, and variable $c_p$. The computational time ($\tau_c$) [min] as a function of the number of CPU parallel processing nodes ($N_p$) is exposed in Figure 9a. Figure 9b shows the total speed-up reached ($\chi$) [× (times)] at each GPU solution as a function of the computational code/hardware. The analysis revealed that commercial code #3 has an optimum number of CPU processing nodes ($N_p$) equal to 7, whereas the other codes are faster for $N_p$ equal to 5. Although for the particular LBW problem simulated here, the variation of $N_p$ often results in little difference in the computational time ($\tau_c$), the optimum values for each software were used in the next analyses. The highest decrease in the computational time was nearly 26.34% for commercial code #3 when switching from $N_p$ equals 3 to 7. Finally, the simulation was executed on an RTX™ 4090 to further enrich the investigation, resulting in a processing time 38.14% faster than that of the RTX™ 3090.

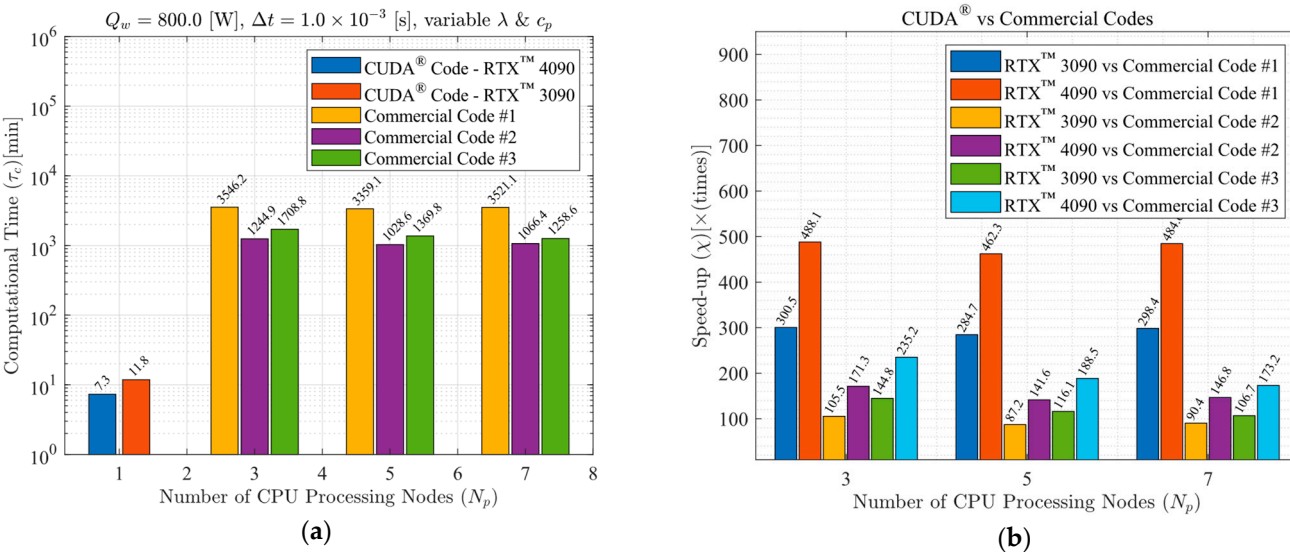

**Figure 9.** Processing performance comparison for 800 [W] LASER power ($Q_w$), $1.0 \times 10^{-3}$ [s] time-step size ($\Delta t$), variable $\lambda$ and variable $c_p$: (**a**) computational time ($\tau_c$) [min] versus number of CPU processing nodes ($N_p$), (**b**) total speed-up reached ($\chi$) [× (times)] as a function of applied code/hardware.

### 3.3. Processing Performance vs. Mesh Size

The GPU solution was investigated in terms of processing performance as a function of mesh refinement. Here, the computational grid sizes $G_4$, $G_5$, and $G_6$ were simulated for 800 [W] LASER power ($Q_w$), $1.0 \times 10^{-3}$ [s] time-step size ($\Delta t$), variable $\lambda$, variable $c_p$, and optimum $N_p$. The overall computational performance of all analyzed codes is presented in Figure 10. Figure 10a illustrates the computational time ($\tau_c$) [min] as a function of the number of mesh total nodes ($N_T$). The total speed-up reached ($\chi$) [× (times)] as a function of computational code/hardware is shown in Figure 10b. It is possible to observe a nearly linear direct proportionality between the number of mesh total nodes ($N_T$) and the required computational time ($\tau_c$) to complete the simulation, with some exceptions noted for commercial code #1. The resultant speed-ups achieved ranged from 75.6 (between CUDA® and commercial code #3, for the 773,490 total nodes mesh) to 1351.2 times (between CUDA® and commercial code #1, for the 773,490 total nodes mesh).

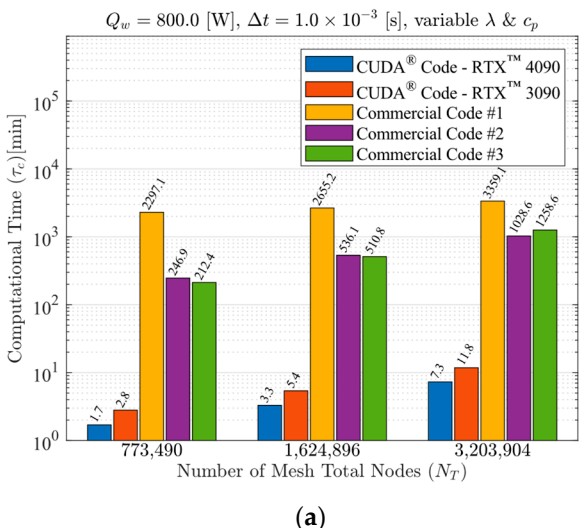
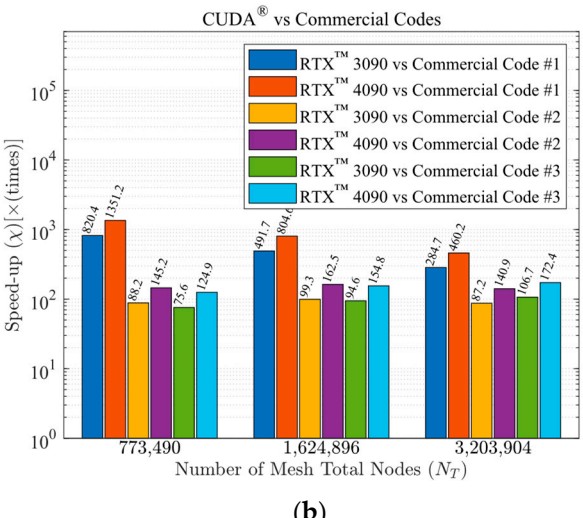

(**a**)                                                                (**b**)

**Figure 10.** Processing performance comparison for 800 [W] LASER power ($Q_w$), $1.0 \times 10^{-3}$ [s] time-step size ($\Delta t$), variable $\lambda$ and variable $c_p$: (**a**) computational time ($\tau_c$) [min] versus number of mesh total nodes ($N_T$), (**b**) total speed-up reached ($\chi$) [× (times)] as a function of applied code/hardware.

### 3.4. Processing Hardware Energy Consumption

An electricity consumption analysis was performed to study the GPU solution in terms of energy efficiency and to ensure no possibility of increasing environmental damages due to the use of GPU simulations. Hence, the numerical solutions had their respective hardware monitored during an LBW simulation case with 800 [W] LASER power ($Q_w$), $1.0 \times 10^{-3}$ [s] time-step size ($\Delta t$), variable $\lambda$, and variable $c_p$. The commercial solutions were rerun in the optimum number of CPU parallel processing nodes ($N_p$) (5, 5, and 7 for commercial codes #1, #2, and #3, respectively). The general energy efficiency of the tested solutions is presented in Figure 11. Figure 11a illustrates the processing hardware electric power ($P_E$) [W] as a function of computational time ($\tau_c$) [s], whereas Figure 11b exposes the processing hardware electricity consumption ($\varepsilon_T$) [kWh] as a function of applied code/hardware. The CUDA® code simulations yield an average of only 5% CPU utilization, and thus, its central processor's power consumption is nearly the same as that of an idle computer. It is important to highlight that the present analysis only accounts for the main processing hardware electricity consumption. Hence, the total computer consumption is significantly higher than the values shown. The investigation evidenced that the GPU solutions required an average of 83.24 times less electrical energy.

The analysis was continued by conducting a cost efficiency study on the energy consumption of the numerical solutions provided by the applied computing methodologies. Here, the investigation was performed based on the electricity rates of where the research was developed (Brazil, city of São José dos Campos) and where it was first exposed (Germany, city of Düsseldorf) in mid-2023. The electricity rates used in the calculations were obtained straight from local energy distribution companies, EDP Brasil and Stadtwerke Düsseldorf. In mid-2023, Brazilian energy rates were governed by the green flag pricing, and hence, the cost per kWh for a commercial low voltage installation (B3 classification) yields the sum between the cost of the electrical energy and the cost for the usage of the distribution system (BRL 0.27614 and BRL 0.37743, respectively). The calculation then totals USD 0.13642 by the commercial conversion rate of BRL 4.791 per dollar (1 August 2023). According to Stadtwerke Düsseldorf, a similar installation would result in EUR 0.3030 per kWh, resulting in a rate of USD 0.33231 by the commercial conversion rate of EUR 1.097 per dollar (1 August 2023). All the monetary quantities were defined here according to the ISO4217 standard [30]. The electricity rates per kilowatt hour may be easily rechecked at each local provider's website. The overall processing hardware electricity cost per simulation ($C_s$) [USD] as a function of computational code/hardware is shown

in Figure 12. The GPU solutions yielded an average cost per simulation 80.57 times lower than the average cost required by the commercial codes (regardless of the country).

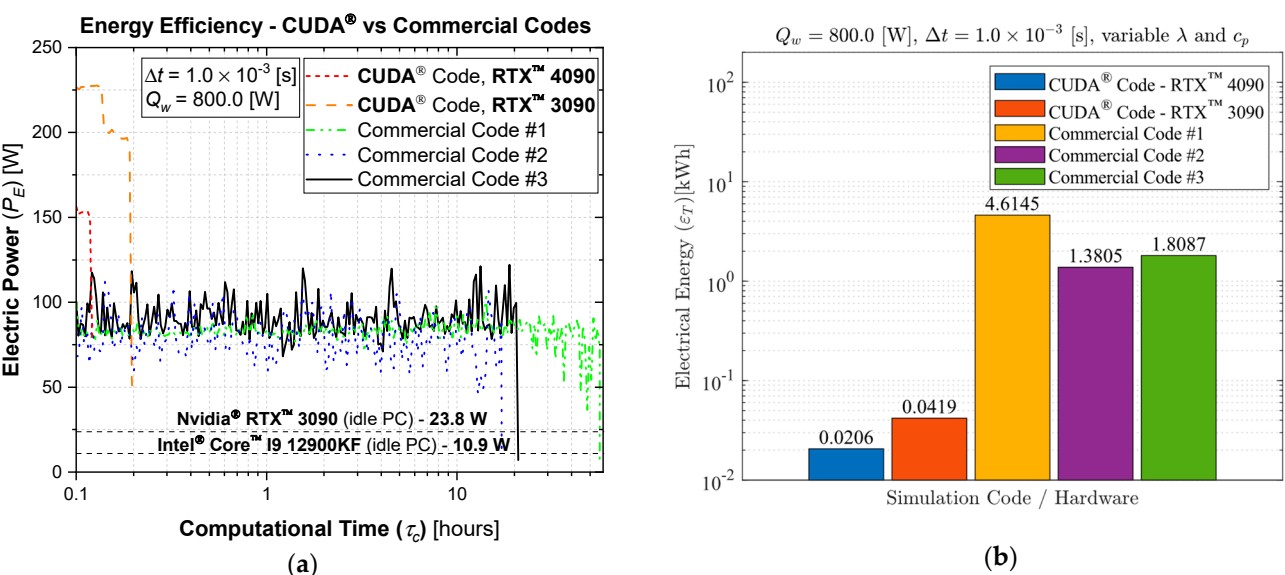

(**a**)                                    (**b**)

**Figure 11.** CUDA® vs. commercial codes energy efficiency comparison for 800 [W] LASER power ($Q_w$), $1.0 \times 10^{-3}$ [s] time-step size ($\Delta t$), variable $\lambda$ and variable $c_p$: (**a**) processing hardware electric power ($P_E$) [W] as a function of computational time ($\tau_c$) [s], (**b**) processing hardware total electricity consumption ($\varepsilon_T$) [kWh] as a function of applied code/hardware.

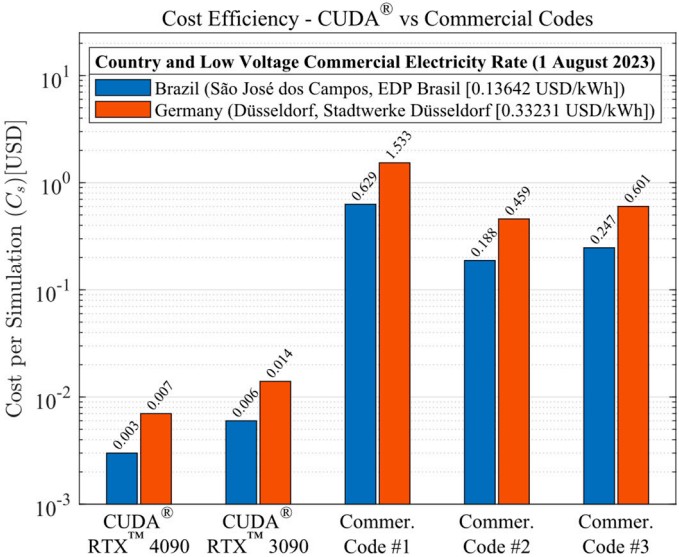

**Figure 12.** Cost per simulation ($C_s$) [USD] as a function computational code/hardware for the Brazilian (São José dos Campos, SP, Brazil) electricity rate [USD] and Germany (Düsseldorf) electricity rate [USD] in mid-2023.

### 3.5. Code Optimization

The GPU solution was investigated in terms of code optimization through a memory usage analysis. This study was performed by comparing the allocated RAM and VRAM at each code during the solution of the LBW problem. The host computer had all apps removed from startup, and the machine was restarted at every new simulation to clean its memory and ensure that each code was running alone. The overall memory performance of the implemented codes is illustrated in Figure 13. The memory usage ($M_u$) as a function of simulated physical phenomenon time (*t*) is shown in letter (a) and represents the system's

total memory usage during simulation (the sum between the operational system and each running code). The average net memory usage ($\overline{M_u\prime}$) was calculated by subtracting the idle PC operational system's memory consumption from each code's average total memory usage value. The CUDA® code was run directly from the Integrated Development Environment (IDE); hence, its RAM memory usage is inevitably summed with the RAM of the GPU simulation code, which is consequently considerably smaller than shown. In spite of that fact, the CUDA® solution still presented an excellent code optimization since its memory usage is by far the lowest among all analyzed codes. The GPU code requires nearly four times less RAM than the average net usage between the three commercial solutions (13.402 GB). The CUDA® solution also offers the advantage of nearly zero memory usage fluctuation, keeping its consumption approximately constant from simulation start to finish.

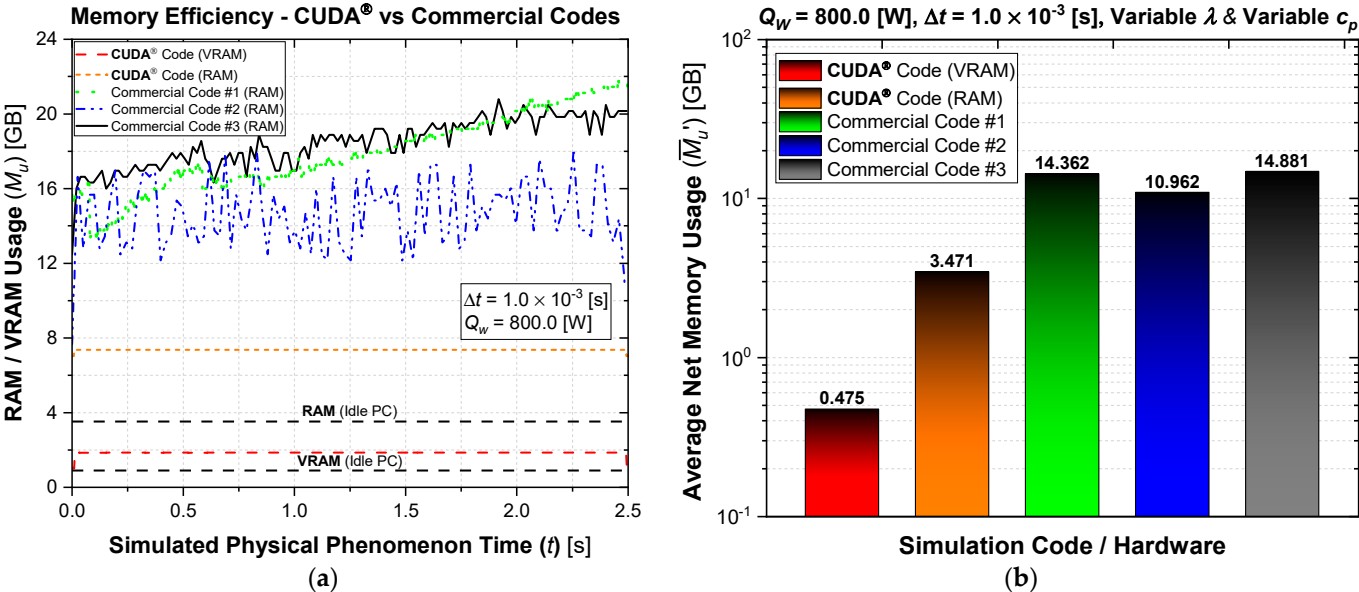

**Figure 13.** Code memory optimization for 800 [W] LASER power ($Q_w$), $1.0 \times 10^{-3}$ [s] time-step size ($\Delta t$), variable $\lambda$ and variable $c_p$: (**a**) memory usage ($M_u$) [GB] as a function of simulated physical phenomenon time ($t$) [s], (**b**) average net memory usage ($\overline{M_u\prime}$) [GB] as a function of simulated code.

## 4. Conclusions

A computational performance analysis of a GPU LASER beam welding implementation using CUDA® was conducted. The applied methodology involved calculating temperature-dependent thermal properties, the temperature-dependent liquid mass fraction function, the coefficients matrix, and the final temperatures of each control volume through multi-thread parallelization. These code functions were executed on the GPU to achieve high-scale parallelism. The CPU was then utilized to coordinate the sequence of execution of all code functions and handle memory management. The results of this implementation were compared to output data from three top-rated commercial codes, assessing accuracy, processing performance, energy consumption, cost efficiency, and code optimization. The GPU solutions demonstrated vast potential in reducing CFD costs and time. The performance investigation yielded speed-ups ranging from 75.6 to 1351.2 times faster than the commercial solutions. This study also demonstrated that each commercial code has an optimum number of CPU parallel processing nodes ($N_p$) that may vary with the type of physics simulated, mesh, number of chip physical cores, and other parameters (for the cases in the present study, $N_p$ = 5, 5, and 7, for commercial codes #1, #2, and #3, respectively). The double precision capability of modern graphics cards was evidenced through their calculations, resulting in an accuracy similar to that of the CPU solutions. Some of the cutting-edge GPU chips have similar or higher Thermal Design Power (TDP) than high-performance CPUs, but end up consuming far less electricity due to the ability to

execute higher parallel processing scaling and thus finishing tasks much faster. As a matter of fact, the investigation revealed that the proposed GPU solutions required an average of 83.24 times less electrical energy in comparison to the commercial codes. In terms of budget, this higher energy efficiency of the GPU solutions resulted in an average cost per simulation 80.57 times lower than the average cost required by the commercial codes (regardless of the country). The in-house code also demonstrated optimized RAM and VRAM usage, averaging 3.86 times less RAM utilization in comparison to the commercial CFD solutions. Lastly, the primary drawbacks of implementing CFD simulations using CUDA® are the heightened coding complexity and the necessity of a CUDA-compatible graphics card. Future work will involve code enhancements through adopting an unstructured multigrid approach.

**Author Contributions:** Conceptualization, E.N. and E.M.; methodology, E.N., E.M. and A.A.; software, E.M., A.A. and E.N.; validation, A.A.; formal analysis, E.N. and A.A.; investigation E.N. and A.A.; resources, E.M.; data curation, E.N. and A.A.; writing—original draft preparation, E.N., A.O. and L.E.S.P.; writing—review and editing, E.N., A.A. and A.O.; visualization, E.N., E.M., A.A., L.E.S.P. and A.O.; supervision, E.M. and L.E.S.P.; project administration, E.M. and L.E.S.P.; funding acquisition, E.M. All authors have read and agreed to the published version of the manuscript.

**Funding:** This research was funded by the Coordenação de Aperfeiçoamento de Pessoal de Nível Superior (CAPES, an agency of the ministry of education of Brazil, under the grant 88887.512576/2020–00), by the Conselho Nacional de Desenvolvimento Científico e Tecnológico (CNPq, an agency of the ministry of Science, Technology, Innovations, and Communications of Brazil, under the grant 402673/2021–2) and by Petróleo Brasileiro S.A. (PETROBRAS, a Brazilian multinational company).

**Data Availability Statement:** The data presented in this study are available on request from the corresponding author.

**Acknowledgments:** The authors would like to thank the Brazilian Ministry of Defense, the Brazilian Airforce (FAB), and the GAP-SJ group for providing food support for the researchers of the Aeronautics Institute of Technology—ITA.

**Conflicts of Interest:** The authors declare no conflict of interest.

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
