# Peer review of "An Implementation of LASER Beam Welding Simulation on Graphics Processing Unit Using CUDA"

_computation, doi:10.3390/computation12040083_

Round 1

Reviewer 1 Report

Comments and Suggestions for Authors

This paper presents a performance comparison between GPU and CPU in CFD by running laser beam welding simulation. The authors implemented a finite volume method (FVM) solver with CUDA-C to run on NVIDIA GPUs. They compared the GPU solution to three commercial CPU-based CFD codes in terms of accuracy, speed, memory usage, and energy efficiency. The GPU code achieved very similar accuracy to the commercial codes while providing a good speedup. 

Overall, the paper clearly presented the method they used and the numerical method appears to be correct. The massive speedups obtained using in-house code with GPU is very impressive, highlighting the immense potential of GPU computing for accelerating CFD simulations. Quantifying the energy and cost savings further strengthens the case. Additionally, the author analyzed the scaling behavior with mesh size and the number of CPU cores for each commercial code provides useful insights.

However, there are a few issues the paper needs to address. 

  • The abstract is a little hard to understand mainly due to language issues. Some editing may be needed to make it more easier to understand and coherent.

  • The introduction section is way too long and lots of the content does not seem necessary or relevant to the topic. For example, the history of numerical methods development, including extrapolation, should just be removed to prevent it from diluting the message, since this paper is not about a new development in numerical methods. Also the various introductions on the effort of trying to use GPU to speed up computation is too verbose. One sentence for each case should be enough. 

  • As the main focus of this paper, the claim about efficiency gain between GPU and CPU is not well founded. The computation efficiency is not just about hardware capacity but also software optimization, especially for complex computations. For this study, the comparison is made between two different codes with different solvers. It is not clear how much of the speed up comes from the hardwas as opposed to the implementation difference between the commercial code and the specifically optimized CUDA code. It's unclear if the 3 commercial CPU codes also used FVM or other discretizations. Comparing FEM or FDM codes may not be fair. Ideally, the comparison should be made with code of the same architecture and same algorithm, but with a different execution engine (on CPU or GPU). If that is not possible, then details should be provided on the CUDA code implementation and the commercial code, such as how the authors parallelized the solver, what optimizations were performed, how the meshes were partitioned across GPU threads vs CPU threads, etc. This would help readers understand how the speedups were achieved.

Based on the points discussed above, instead of trying to make a general performance comparison of CFD between GPU and CPU, while using the simulation of LBW as a case study, I would strongly suggest authors to make the implementation of LBW simulation code on GPU the main selling point of this paper. You could use the performance comparison between the new implementation on GPU vs commercial code on CPU to justify your new approach. Consequently, the title should be changed to something like “ An Implementation of Laser Beam Welding simulation on GPU Using XXX”. The abstract and content should also be adapted. I believe it would be an excellent paper if you make such an adaptation.

Comments on the Quality of English Language

There are very few english issues detected in this paper except the abstract, which need some language editing.

  • The first sentence is not quite right (or is a little ambiguous). SIMD came out in the 1990s and multi-core CPUs came out even earlier than programmable GPUs.

  • The expression of the sentence at line 13 is hard to understand. It could be rephrased as "A computational performance comparison between engineering codes run on GPU and CPU"

  • In line 14, it should be “this analysis aimed at …”

  • In line 21, the sentence could be "The results demonstrated that GPU and CPU processing achieve similar precision"

Author Response

Dear Reviewer 1,

Attached you will find the PDF containing the responses to all the points raised in Review 1 of manuscript computation-2942879. I sincerely thank you for dedicating your time to review this work. I eagerly await your feedback.

Best Regards,

Dr. Ernandes Nascimento

Reviewer 2 Report

Comments and Suggestions for Authors

This work conducted a computational performance analysis between GPU and CPU-based codes. I have the following questions:

(1) Please add P1-P4 in figure 2.

(2) What are commercial code #1~#3?

(3) What's the point to compare CUDA with commercial code? Does it mean CUDA more accurate?

Comments on the Quality of English Language

Minor editing of English language required

Author Response

Dear Reviewer 2,

Attached, you will find the PDF containing the responses to all the points raised in Review 1 of manuscript computation-2942879. I sincerely thank you for dedicating your time to review this work. I eagerly await your feedback.

Best Regards,

Dr. Ernandes Nascimento

Round 2

Reviewer 1 Report

Comments and Suggestions for Authors

I appreciate the author's thoughtful response to the suggestions proposed in the first review, particularly regarding the title update to shift the focus of the paper from making a general conclusion about GPU acceleration in CFD simulations to emphasizing the specific implementation described in this work. While the title has been revised, the conclusion section may need to be slightly updated to align with this change. In the conclusion section, the author may briefly summarize the method and implementation of the GPU simulation code before went ahead to discuss the performance comparison and highlight the potential advantages of CFD simulations on GPU over CPU.

Regarding the author's response to the comments about comparing the in-house GPU solution with commercial code, I agree that this comparison can be meaningful by itself as a showing case in achieving speedup using GPU. However, my concern was that using a special case with many unknown variables may not be sufficient to support a general claim about GPU and CPU performance in CFD, which was the main focus of the previous manuscript. That was also the reason why I suggested the author update the title to shift the focus in the first place.

Author Response

Dear Reviewer 1,

Attached, you will find the summary of changes for the second review of the manuscript. I look forward to hearing from you soon.

Best regards,

Dr. Ernandes Nascimento
